# Position: Video LLMs Must Not Ignore the Pixel Dynamics in Plain Sight

**Shayda Moezzi** [1]  **Umer Saleem** [1]  **Andong Deng** [2]  **Chen Chen** [2]  **Sarah Ostadabbas** [1]

## Abstract

The essence of video lies in pixel dynamics: motion, state transitions, and the flow of visual information across frames. Video Large Language Models (LLMs) have rapidly become the dominant paradigm for video understanding in computer vision, sophisticated multimodal reasoning over complex, long-form visual streams. In this position paper, we argue that recent progress in video understanding is measured by benchmarks and protocols that can be solved without reliably perceiving spatiotemporal evidence, rewarding language-driven plausibility over video-grounded inference. We identify two coupled failure modes that consistently emerge across recent Video LLM evaluations: (i) static-cue dominance, where appearance and context outweigh spatiotemporal evidence, and (ii) prior-driven temporal hallucination, where learned regularities fill in temporal and causal structure when dynamics are subtle or counterintuitive. We synthesize recent diagnostic probes that expose these failure modes into a call to action for the community: to re-center video understanding on what a video uniquely contains, namely, dynamic evidence that unfolds over time, by enforcing spatiotemporal grounding in both models and benchmarks, before the pixel dynamics are lost in plain sight. Additional visual examples and project materials are available on the project page.[1]

## 1. Introduction

**This position paper argues that current progress in Video LLMs is constrained by static-cue dominance, in** which appearance and contextual signals substitute for spatiotemporal evidence, and prior-driven temporal hallucination, where learned regularities impose temporal and causal structure when observed dynamics are subtle or counterintuitive.

Video is inherently temporal: the meaning of an event often lies not in what appears in a single moment, but in how states change, interact, and unfold over time (Huang et al., 2018). Human perception relies fundamentally on motion and temporal continuity to understand a dynamic world (Bill et al., 2022); motion separates figure from background, reveals three-dimensional structure (Beer et al., 2009), and supports prediction of future states. Video makes this dynamic structure explicit by encoding motion, interaction,

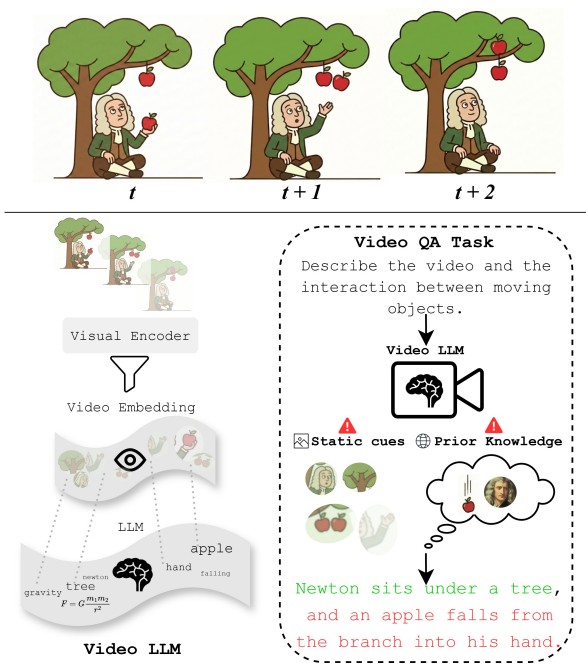

*Figure 1.* **Overview of Video LLM Failure Modes.** We use this example to illustrate two failure modes of Video LLMs. Static-Cue Dominance: salient appearance/context cues overshadow low-signal but decisive pixel dynamics, causing the upward throw to be underweighted or missed. Prior-Driven Temporal Hallucination: the model's learned event priors override the observed motion and complete the most likely script (here, hallucinating that the apple falls from the tree despite the video showing an upward toss).

[1]Electrical and Computer Engineering Department, Northeastern University, Boston, MA, USA. [2]Department of Computer Science, University of Central Florida, Orlando, FL, USA. Correspondence to: Sarah Ostadabbas <s.ostadabbas@northeastern.edu>.

*Proceedings of the $43^{rd}$ International Conference on Machine Learning*, Seoul, South Korea. PMLR 306, 2026. Copyright 2026 by the author(s).

[1]https://ostadabbas.github.io/Lost-in-Plain-Sight/

and causality directly in pixel-level changes across time.

Video Large Language Models (LLMs) (Comanici et al., 2025; Achiam et al., 2023; Bai et al., 2025a; Wang et al., 2025; Li et al., 2024a) have rapidly become the dominant paradigm for video understanding, complex multimodal reasoning over continuous visual streams by coupling visual embedding to an LLM. This paradigm has driven rapid progress in video question-answering (Grauman et al., 2022; Krishna et al., 2017) and captioning (Yang et al., 2023; Sun et al., 2019), yielding impressive benchmark scores that suggest temporal reasoning capabilities. However, this impression is misleading. We argue that the field is suffering from a blind spot: much of today's progress is quantified by benchmarks and protocols that can be solved without reliably using spatiotemporal evidence, allowing models to succeed via appearance, context, and learned regularities rather than by tracking state evolution (Cores et al., 2025; Varma et al., 2025; Feng et al., 2025a).

To understand the fragility of current Video LLMs, we must look to the recent history of Vision-Language Models (VLMs) on image domain, which offers a stark cautionary tale. Research has repeatedly discovered that high benchmark scores often mask fundamental perceptual deficits. Seminal diagnostic studies, such as Winoground (Thrush et al., 2022) and work by (Yuksekgonul et al., 2023), demonstrated that VLMs frequently behave as "bag-of-words" models. They excel at recognizing object co-occurrences (e.g., identifiying "dog" and "grass") but fail catastrophically at compositional reasoning. As highlighted in (Chen et al., 2024b), standard evaluation metrics often conflate text matching with visual understanding, rewarding models for outputting keywords regardless of their visual grounding. Further analysis using explainability methods revealed that models often focus on irrelevant background pixels or exploit "multimodal answer leakage", where the question phrasing itself narrows the solution space so significantly that the visual input becomes redundant (Li et al., 2025b).

This reliance on shortcuts has not only extended to the video domain but has intensified. In videos, the "bag-of-words" failure manifests as "bag-of-frames". Because video understanding benchmarks are often saturated with overly simple samples (Cores et al., 2025), models can leverage strong language priors to solve videos without processing the temporal signal (Han et al., 2025). Consequently, we face a fundamental epistemic uncertainty: are current Video LLMs truly understanding temporal structure, or are they merely remembering the most likely linguistic narrative while ignoring the pixel dynamics?

Together, these patterns misorder the evidence hierarchy of video understanding, allowing models to succeed without explicitly tracking state evolution over time. We therefore call for a shift in practice: temporal and causal claims must be grounded in observed pixel dynamics, and both architectures and benchmarks should treat spatiotemporal evidence as a first-class requirement–before normalizing models that achieve high performance while overlooking what is in plain sight. Figure 1 illustrates the core failure modes in current architectures where semantic cues and language model priors blind the model to the visual evidence of an upward toss with the canonical narrative of a falling apple.

This paper synthesizes recent diagnostic evidence into a unified failure taxonomy for Video LLMs. It contributes conceptual unification and prescriptive evaluation principles that enforce spatiotemporal grounding. Using this taxonomy, we sharpen two recurring failure modes and motivate protocols in which temporal and causal claims are verifiable from the video. Finally, we consider alternative views–that static semantics, world knowledge, and "good-enough" temporal cues may suffice for many applications–but argue that video understanding must remain verifiable from pixel dynamics before the evidence is **lost in plain sight**.

## 2. The Illusion of Success: Benchmark Alignment and Architectural Biases

With the rapid progress on architecture design (Li et al., 2024a; Yang et al., 2025a; Wang et al., 2025), training data construction (Grauman et al., 2022; Wang et al., 2023; Bain et al., 2021; Xue et al., 2022), and training recipe development (Cheng et al., 2024; Wang et al., 2024; Feng et al., 2025b), current Video LLMs have achieve remarkable performance on recent video benchmarks. For example, on standard benchmarks such as Video-MME (Fu et al., 2025), MVBench (Li et al., 2024b), EgoSchema (Mangalam et al., 2023), and LongVideoBench (Wu et al., 2024), both proprietary models (Comanici et al., 2025; Achiam et al., 2023) and open-source models (Yang et al., 2025a; Wang et al., 2025; Li et al., 2024a; Cheng et al., 2024) are achieving high performance. However, when exposed to rudimentary temporal understanding and movement perception, such as distinguishing an object moving from left to right or determining if a door is opening or closing, indicated by recent studies (Hong et al., 2025; Cores et al., 2025; Tu et al., 2025; Tang et al., 2024), these powerful models disappointingly exhibit catastrophic failures. The fragility of current Video LLMs is best illustrated by the physically invalid Newton's cradle in Figure 2. While a human observer would immediately understand the "impossible" nature of the stationary end-ball, the model overrides this dynamic evidence in favor of a fluent physics-compliant narrative. This reveals a worrying fact that the apparent success of Video LLMs reflects how well they satisfy benchmark requirements, not necessarily how well they perceive over temporal dynamics.

## 2.1. A Benchmark Perspective

To understand why these models succeed so consistently on these benchmarks while failing at elementary temporal perception, it is necessary to investigate how current benchmarks actually reward. If a video understanding benchmark can be solved without seeing the video, it is not measuring video understanding; it is measuring language reasoning. Recent analysis by (Feng et al., 2025a) reveals that widely adopted benchmarks (Fu et al., 2025; Mangalam et al., 2023; Xiao et al., 2021; Zhou et al., 2025a) contain a staggering proportion of questions that are "LLM-Answerable", meaning a blind language model can correctly guess the answer based solely on the questions and options. Furthermore, they found that an additional subset of questions in NextQA and EgoSchema are "Semantic-Only", solvable even when video frames are randomly shuffled, stripping away all causal and temporal structure. This confirms a critical vulnerability: our current progress metrics are largely insensitive to time. (Cores et al., 2025) further demonstrates this with TVBench, showing that while SOTA models excel on MVBench and ActivityNet-QA (Yu et al., 2019), their performance barely changes if the video frames are shuffled. This implies that for current models, a video is effectively a bag of frames, where temporal order is largely ignored.

As a consequence, in the absence of strong temporal evidence, models learn to prioritize language priors over pixel dynamics in order to align with benchmark objectives. Recent work in (Han et al., 2025) suggests that LLMs acquire implicit visual knowledge purely from text pre-training (e.g., knowing that "glass" implies "breaking" if dropped). In Video LLMs, this manifests as sycophancy, the tendency to prioritize user-suggested narratives over contradictory visual facts, and hallucination, the invention of likely (but unseen) details to preserve narrative coherence. (Zhou et al., 2025b) exposes that Video LLMs frequently parrot user inputs or high-probability narrative completions, even when the visual evidence directly contradicts them; strong textual priors and weak temporal signals lead models to hallucinate expected actions rather than perceive actual effects.

## 2.2. An Architecture Perspective

The aforementioned behaviors are also closely tied to the design paradigm of current Video LLMs and the way visual information is integrated into language models. Basically, current architectures generally consist of a visual encoder, a fusion module, and an LLM backbone. This design is effective for scaling, but it also systematically compresses or defers temporal information, biasing the system toward language-driven inference. As discussed in (Ding & Wang, 2025), current visual encoders fundamentally bottlenecks the key spatiotemporal information in videos. Image-based encoders from the CLIP family (Radford et al., 2021; Sun et al.,

2023; Zhai et al., 2023) dominate current designs due to their strong semantic representations learned through large-scale contrastive training; however, their frame-independent nature prevents them from directly modeling temporal relationships. Video-native encoders (Bertasius et al., 2021; Tong et al., 2022) partially alleviate this limitation by incorporating spatiotemporal attention, but their pretraining on categorical action labels constrains their ability to capture fine-grained, continuous pixel dynamics. The fusion module determines how visual information is integrated into language generation. Simple linear projections (Liu et al., 2023) offer efficiency and ease of integration by treating visual tokens as static embeddings and shifting temporal reasoning to the LLM, while Q-Former (Li et al., 2023) compress visual inputs through a small set of learnable queries, enabling longer context handling at the cost of discarding fine-grained temporal variation. For example, analysis by (Jiang et al., 2025) reveals that standard projection layers preserve noun-like object existence and their most probable associated behaviors, while failing to encode the verb-like, continuous dynamics that emerge only through transitions over time. While a growing body of work has introduced temporal-specific mechanisms to mitigate this issue (Nie et al., 2024; Cheng et al., 2024), such designs remain auxiliary to the dominant fusion paradigm, leaving temporal dynamics weakly represented between vision and language.

⑦ **Describe the motion of the balls in the video.**

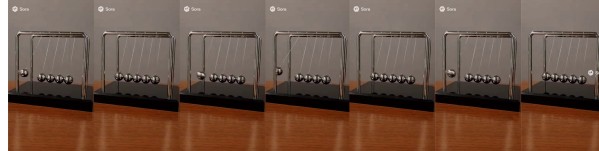

The video captures the classic mesmerizing motion of a Newton's Cradle. The action centers on the outer two balls while the three in the middle remain largely stationary. The ball on the far left swings down and strikes the stationary line of balls. Upon impact, the energy travels almost invisibly through the three middle balls, which barely move. The momentum is transferred to the ball on the far right, causing it to swing outward and upward to a height nearly equal to the first ball's release point. Gravity pulls the right ball back down to strike the group, reversing the flow of energy and sending the left ball swinging out again. This back-and-forth rhythmic cycle repeats continuously throughout the clip.

*Figure 2.* **Prior-Driven Hallucination on a Physics Violation.** In this Sora-generated Newton's Cradle video, the rightmost ball remains stationary after impact (violating momentum transfer). Gemini-2.5-Pro (Comanici et al., 2025) nevertheless describes the canonical outcome (right ball swings), with hallucinated claims highlighted in red.

Consequently, as the reasoning engine, the LLM backbone leverages extensive pretrained language knowledge to interpret and generate responses conditioned on visual tokens. As previously discussed, when these tokens carry limited spatiotemporal detail due to limited modeling capability, the LLM naturally resorts to inferring event structure and

causality. This inference relies on the sequence modeling ability and causal attention mechanism (Vaswani et al., 2017; Brown et al., 2020; Ding & Wang, 2025), which imposes an ordering over visual tokens but do not actually encode continuous temporal dynamics. For instance, (Shi et al., 2025) provides evidence that even when temporal information is injected via positional embeddings, Video LLMs still rely on the causal attention mask to infer sequence structure. Therefore, under benchmark settings where video is only viewed as a bag of frames, such language-driven inference is often sufficient to produce plausible answers.

The convergence of these systematic blind spots creates a precarious trajectory for the field, which brings us to the core of our position: we are optimizing for systems that are articulate but ignore dynamics. Re-grounding video understanding, therefore, requires confronting the concrete failure modes that emerge from this paradigm. In the next section, we examine these failures in detail and identify two recurring patterns that characterize current Video LLM behavior in video understanding.

## 3. Failure Modes of Current Video LLMs

Video LLMs can appear temporally competent while systematically underusing the very signal that makes video distinct: dynamic evolution in pixels. By *pixel dynamics*, we mean the information carried by transitions between frames: object motion, direction, speed, state change, event order, and causal interaction. This is the part of the visual signal that cannot be resolved from a single frame alone and should not be recoverable purely from language priors. To make this concrete, we characterize the dominant pathologies reported across recent diagnostic probes (Table 2) as two failure modes.

**How this differs from classical video bias**    Static and scene biases are not new in video understanding. Classical action recognition also suffered from contextual shortcuts, but many video-first tasks made temporal grounding directly consequential. For example, tracking could fail through trajectory drift, segmentation through low mask overlap, and action localization through incorrect temporal boundaries. In contrast, the dominant Video LLM interface maps video to fluent language, often evaluated through answer correctness or semantic plausibility. This makes the same underlying bias structurally easier to hide. Our position is therefore not that Video LLMs invented static-cue reliance, but that current Video LLM protocols normalize it by making temporal grounding optional rather than mandatory.

### 3.1. Failure Mode 1: Static-Cue Dominance

**The structural origin: frame-centric inheritance**    The prevailing paradigm in Video LLMs is still an inheritance of image-language modeling: video is treated as a set of sampled frames rather than a continuous spatiotemporal signal. In most pipelines, frames are first mapped to image embeddings via a pretrained vision encoder (Sun et al., 2023; Zhai et al., 2023), and only then passed through a comparatively shallow fusion module, e.g., a linear layer (Liu et al., 2023), limited cross-attention (Li et al., 2023), etc. This design effectively induces a *bag-of-frames* representation: the model can access what is present, but is weakly constrained to represent what *changes*. The result is a systematic bias toward static recognition, where "understanding" is often reducible to matching objects and scenes to high-probability action labels. The same cautionary pattern has been documented in image understanding tasks with VLMs (Tong et al., 2024): strong language coupling and dataset regularities can yield seemingly competent models that are weakly grounded in visual evidence. Video LLMs inherit this paradigm, but the cost is amplified: what was an image-grounding failure becomes a temporal one, where state transitions, directionality, and event evolution are treated as optional signals.

**Static features trigger blind temporal decisions**    Multiple studies show that Video LLMs hinge on spurious static features when those features are predictive under the training distribution. Shortcut analyses in TRoVe (Varma et al., 2025) demonstrate that Video LLMs associate static context with action labels, leading to high confidence predictions even when motion evidence is absent or contradictory. Such reliance on appearance over dynamics degrades robustness under distribution shifts (Li et al., 2022). This failure is most evident in Minimal Video Pairs evaluations (Krojer et al., 2025): when appearance is held fixed and only the temporal trajectory varies, even proprietary models like Gemini-1.5 Pro and GPT-4o perform below chance.

*Table 1.* **Motivating Motion Probes on Simulated Physics.** Accuracy (%) on 3 s collision videos from AVoE (Dasgupta et al., 2021) using 16 sampled frames and a Binary Yes/No task (*Does the left/right object change direction after collision?*). We evaluate 50 expected and 50 surprising (VoE) clips.

| Model | Input Frames | Expected | Surprising |
|---|---|---|---|
| Random Baseline | – | 50.0 | 50.0 |
| GPT-4o (Achiam et al., 2023) | 16 | 59.0 | 47.0 |
| Gemini-2.5-Pro (Comanici et al., 2025) | 16 | 57.0 | 63.0 |
| InternVL3-8B (Wang et al., 2025) | 16 | 57.0 | 50.0 |
| VideoLLaMA3-7B (Zhang et al., 2025) | 16 | 57.0 | 46.0 |
| Qwen2.5-VL-7B (Bai et al., 2025a) | 16 | 60.0 | 62.0 |

This failure is not limited to discrete state changes; it also manifests as a failure to perceive continuous flow that is only defined between frames. The Escalator Problem (Zhang, 2025) formalizes this as implicit motion blindness: directionality is not recoverable from any single frame and requires integrating change between frames. Current models frequently revert to static scene descriptions, express uncertainty about movement, or guess based on contextual

*Table 2.* Synthesis of recent diagnostic probes and the spatiotemporal failure modes they expose in Video LLMs. Each entry maps a diagnostic methodology to the two core failure modes of Video LLMs: **(i) Static-Cue Dominance** and **(ii) Prior-Driven Temporal Hallucination**. Rows are ordered by failure-mode alignment: static-only → prior-only → coupled.

| Benchmark | Probe Mechanism | Exposed Blind Spot | Failure mode | |
|---|---|---|:---:|:---:|
| | | | *Static* | *Prior* |
| **Static-Cue Dominance** | | | | |
| **TRoVe** (Varma et al., 2025) | Mines recurring static visual clusters in benchmark validation data and measures models' reliance on static features. | Models bypass temporal dynamics, predicting actions based on objects rather than motion. | ✓ | |
| **Vinoground** (Zhang et al., 2024) | Pairs natural videos with captions containing identical words but opposing temporal order. | Models fail to distinguish events with identical static semantics but distinct timelines, indicating a collapse of temporal order into static co-occurrence. | ✓ | |
| **MESH** (Yang et al., 2025b) | QA designed with bottom-up cognitive path (setting → action) with distractors that are contextually plausible but visually absent. | Models select context-consistent traps that fit scene context | ✓ | |
| **UTD** (Shvetsova et al., 2025) | De-biases benchmark QA by removing items solvable via single-frame objects/attributes. | Model performance collapses on de-biased test splits, indicating that many performance gains come from appearance shortcuts, not temporal evidence use. | ✓ | |
| **TempCompass** (Liu et al., 2024a) | Benchmark of video pairs sharing identical static content but differing in events and action over time. | Perfomance drops on conflicting pairs reveal reliance on static semantics; models fail to distinguish events that look alike but move differently. | ✓ | |
| **MotionBench** (Hong et al., 2025) | Curated questions explicitly targeting motion cues and temporal change rather than static recognition. | Models often capture scene semantics but fail on fine-grained motion understanding, indicating that dynamics are weakly represented relative to appearance cues. | ✓ | |
| **FAVOR-Bench** (Tu et al., 2025) | Probes subtle motion properties (e.g., amplitude, frequency, manner). | Models can name an action class yet fail on how it unfolds, indicating missing motion-level representations. | ✓ | |
| **Prior-Driven Temporal Hallucination** | | | | |
| **VideoHallucer** (Li et al., 2025a) | Pairs positive (ground-truth) queries with negative (hallucinated) queries to test discrimination consistency. | Models consistently affirm plausible but absent details, with more pronounced hallucinations in high-parameter models. | | ✓ |
| **UNSCENE** (Bae et al., 2025) | Probes reliance on priors via incongruent action-scene pairs (e.g., boxing in a library) and actor-free scenes. | Models rely on scene priors over pixel evidence, hallucinating actions from backgrounds. | | ✓ |
| **CounterVid** (Poppi et al., 2026) | Synthesize AI-generated video pairs sharing identical start-frames (anchors) but diverging in action or temporal order | Models hallucinate actions/orders based on scene associations. | | ✓ |
| **VideoHallu** (Li et al., 2025d) | Synthesize AI-generated videos of impossible events, testing is models detect violations. | Models fail to notice physics violations and answer based on how the world "should" work. | | ✓ |
| **VERHallu** (Zhang et al., 2026) | Tests causal, temporal, and subevent relations in videos that defy typical scripts. | Models recognize isolated key events but hallucinate the links (cause/effect) between them based on scene context, failing to track actual dynamic transition. | | ✓ |
| **NOAH** (Lee et al., 2025) | Inserts controlled event clips into videos at varying semantic similarities and positions to test robustness against narrative disruption. | Models hallucinate transitions or omit incompatible events to force a coherent storyline. | | ✓ |
| **Static + Prior** | | | | |
| **MVP** (Krojer et al., 2025) | Requires correct answers on paired videos with near-identical scenes but opposite temporal outcomes (e.g., open vs. close) to penalize guessing. | Models fail to distinguish minimal temporal changes in videos with near-identical appearance; indicates missing representations of subtle trajectories. | ✓ | ✓ |
| **TVBench** (Cores et al., 2025) | Compares performance on normal vs. shuffled/reversed inputs to quantify reliance on order-agnostic static cues. | Negligible performance drops on shuffled inputs reveal that models and existing benchmarks treat videos as unordered "bags of frames", ignoring sequential dynamics. | ✓ | ✓ |
| **VBenchComp** (Feng et al., 2025a) | Categorizes existing benchmark QA into LLM-answerable and Semantic (shuffled-invariant) buckets to isolate true temporal dependencies | Models often maintain performance even when frames are shuffled, revealing high benchmark scores stem from static cues and priors. | ✓ | ✓ |
| **HERBench** (Ben-Ami et al., 2025) | Designs QA that demand "high-evidential requirement" across at least 3 frames in the video; uses oracle frames to disentangle retrieval failures. | Models fail to aggregate dispersed cues over time; even when explicitly provided, revert to salient cues in a single frame. | ✓ | ✓ |
| **REXTIME** (Chen et al., 2024a) | QA-IoU metric enforces non-overlapping query/answer spans across distant segments. | Performance drops when correct answers require linking separated events, exposing shallow temporal memory/credit assignment. | ✓ | ✓ |
| **MHBench** (Kong et al., 2025) | Video triplets original vs. antonym vs. incomplete actions with shared objects; tests action existence and polarity. | Models hallucinate actions from object presence, yielding temporally incorrect claims even when motion semantics are adversarial. | ✓ | ✓ |
| **Dr.V-Bench** (Luo et al., 2025) | Evaluates perceptive, temporal, and cognitive hallucinations with fine-grained spatial-temporal grounding. | Errors range from missed relations to fabricated temporal/cognitive explanations, motivating staged verification over raw generation. | ✓ | ✓ |

priors. This pattern generalizes to other continuous-flow settings (e.g., revolving doors, crowd flow, water currents), indicating that the current visual encoding mechanisms used in Video LLMs inherit systematic blind spots precisely where temporal evidence is most essential. To move beyond aggregate benchmarks, we conducted a small experiment using examples from the AVoE dataset (Dasgupta et al., 2021), which isolate motion primitives from complex

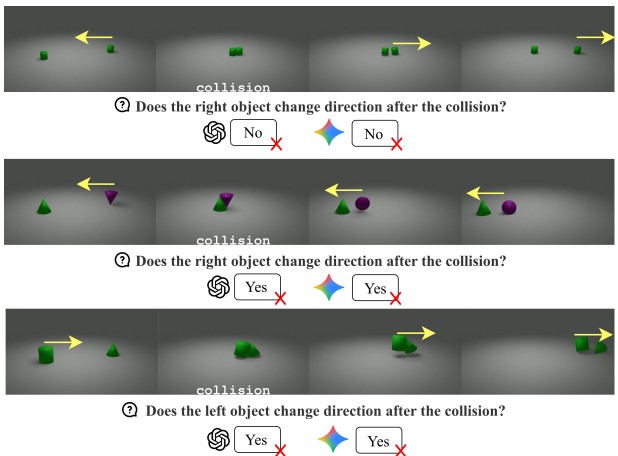

*Figure 3.* Example 3 s videos from AVoE ([Dasgupta et al., 2021](#)) (16 uniformly sampled frames as input) with a binary QA prompt: *Does the left/right object change direction after collision?* (Yes/No). Yellow arrows are overlays for visualization only.

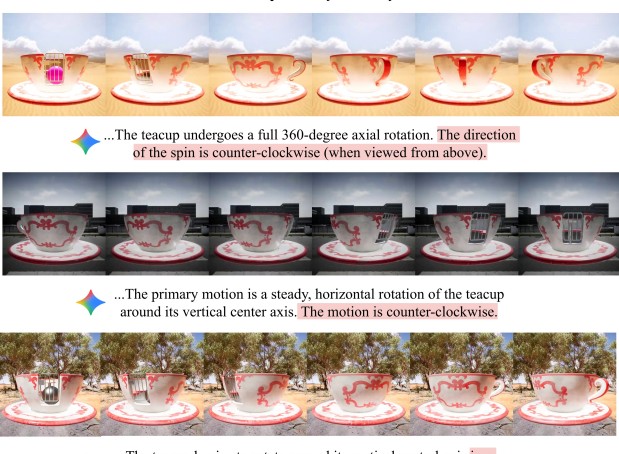

*Figure 4.* **Visualizing Motion Blindness** Gemini-2.5-Pro ([Comanici et al., 2025](#)) is prompted to describe scene dynamics on IntPhys2 teacup-rotation clips ([Bordes et al., 2025](#)), sampled at 8 fps. Although the teacups rotate clockwise from the camera perspective, the model repeatedly reports counter-clockwise rotation; incorrect phrases are highlighted in red

semantic backgrounds. By tasking models with a simple binary choice, determining if an object changes direction after a collision, we can expose the degree to which they understand and track a simple motion trajectory, in a setting void of context. The results in Table 1 motivate our position. Even frontier models like GPT-4o ([Achiam et al., 2023](#)) and Gemini 2.5 Pro ([Comanici et al., 2025](#)) perform near-random in such a simple setting. This failure is visualized in Figure 3. While this specific probe is limited in scope, it highlights a fundamental inability to distinguish specific motion primitives even in simple scenes. These initial findings are further validated and expanded upon by

the broader suite of diagnostic probes synthesized in Table 2, which confirm that this motion blindness is a systemic architectural issue rather than a dataset-specific quirk. As a motivating qualitative probe, we evaluate Gemini-2.5-Pro on IntPhys2 teacup-rotation clips ([Bordes et al., 2025](#)) using prompts that require identifying the direction of rotation. Figure 4 shows that the model repeatedly misclassifies the rotation direction (i.e., fails to recover the correct temporal directionality from the video). Although this probe is small-scale and intended for visualization rather than statistical claims, it is consistent with the broader failure patterns we analyze throughout the paper.

**Motion-focused diagnostics expose static-cue dependence** Recent benchmarks designed to disrupt appearance-first strategies provide the strongest evidence for this failure mode ([Tu et al., 2025](#); [Hong et al., 2025](#)). These diagnostics move beyond aggregate accuracy to verify if predictions remain sensitive to motion when static sufficiency is removed. TempCompass ([Liu et al., 2024b](#)) exemplifies this shift by isolating temporal factors, such as direction and speed, using conflicting cases where identical static content requires different temporal answers. Their results demonstrate that while models leverage static visual cues to identify actions, they struggle with motion blindness when confronted with conflicting videos designed to eliminate single-frame bias and language priors. Further insight comes from motion-centric benchmark ([Hong et al., 2025](#); [Tu et al., 2025](#)) which decomposes comprehension into motion-focused queries. They find that fine-grained motion represents the largest share of these failures; even in short clips (0-4s), models perform below random at 11-14% (where random performance is 25%). This suggests that current deficits stem not from long-range context issues, but from a fundamental inability to distinguish specific motion primitives even when content is limited.

### 3.2. Failure Mode 2: Prior-driven Temporal Hallucination

When the visual evidence in a video is subtle or counter-intuitive, Video LLMs often do not default to uncertainty. Instead, they hallucinate a false reality. We term this failure mode Prior-driven Temporal Hallucination: the system substitutes canonical event scripts for evidence-based state tracking, producing fluent but incorrect descriptions of events in a video.

**Events Hallucinated from Entity Co-occurrences** Models often describe actions simply because the relevant entities are present. In the bottom-left example of Figure 5, Gemini perceives a pendulum and spring system; rather than recognizing the fact that the right pendulum never moves, it hallucinates a rightward motion driven by the "stored energy" due to the movement of the left pendulum and the

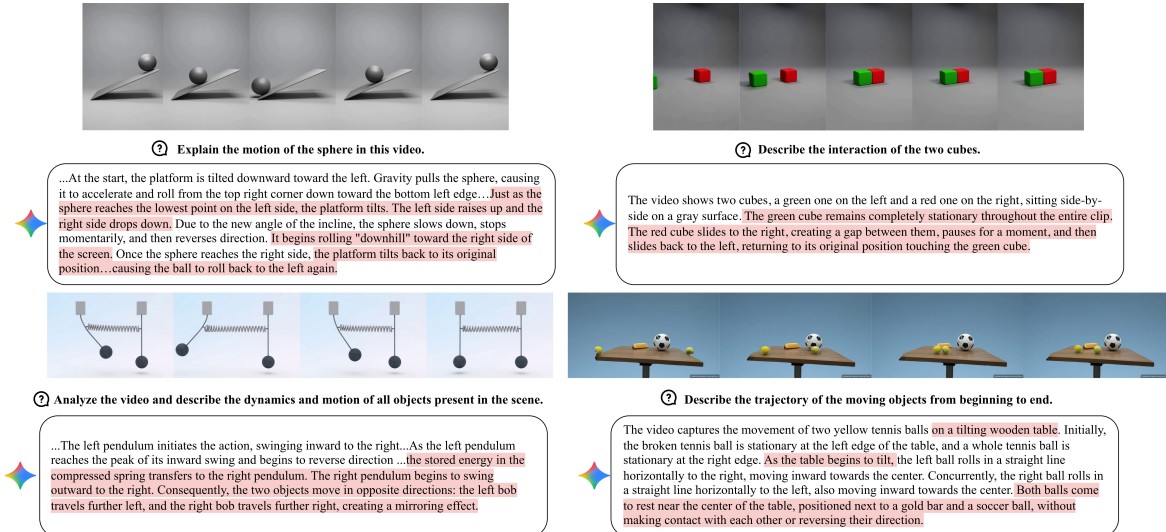

*Figure 5.* **Isolating Dynamics in Plain Sight** We demonstrate critical failures in Gemini-2.5-Pro using synthetic videos generated via Grok Imagine (Top Row, Bottom-Left) and the QuantiPhy dataset (Li et al., 2025c) (Bottom-Right). All videos were sampled at 8 fps. *(Top-Left)* The model hallucinates environmental change to explain the sphere's ascent on a static incline in the absence of outside forces. *(Top-Right)* The model falsely identifies the actively moving green cube as stationary. *(Bottom-Left)* The model fabricates a trajectory in the right pendulum that never manifests in the pixel dynamics. *(Bottom-Right)* The model invents an unseen physical catalyst to describe motion, and additionally, never identifies the collision between the tennis balls.

spring. A deeper analysis in (Kong et al., 2025) exposes this vulnerability through adversarial triples (sequences containing original, reversed, and antonym actions with shared objects) and finds that models frequently affirm actions from static co-occurrence cues even when the temporal polarity contradicts the claim. This suggests that the video understanding task is shifting towards a reasoning task where static semantic cues trigger a pre-computed motion script. (Ben-Ami et al., 2025) shows that even when provided with oracle frames containing necessary information, models rely disproportionately on a single salient snapshot rather than aggregating spatiotemporal evidence. Under this deficit, Video LLMs fill missing temporal structure using language priors, rendering them blind to the actual state evolution.

**Counterfactual Videos** Prior dominance becomes most visible when the video dynamics violate the norm. Benchmarks with counterfactual videos (Bai et al., 2025b; Poppi et al., 2026) and matched-content but altered temporal outcomes (Yang et al., 2025b) show that models often output canonical but false outcomes over visually supported atypical outcomes. Under temporal gaps, models similarly fabricate a plausible description over the ground truth. The reliance on priors creates a perverse scaling law as models become more articulate, they become more competent fabricators. Larger language backbones often exacerbate this problem by enabling the insertion of plausible but invisible details to satisfy narrative expectations (Li et al., 2025a). Analysis reveals that while parameter scaling improves object detection in scenes, it simultaneously degrades the model's ability to refrain from over-explaining the scene

with unseen details. In the bottom-right of Figure 5, Gemini attempts to account for the tennis balls moving inward by describing the table as tilting, a structural fabrication used to provide an explanation for a trajectory the model cannot perceptually ground.

Furthermore, Video LLMs display a fundamental intolerance for discontinuity; when faced with temporal gaps or disjointed events, they do not report the fragmentation but actively smooth it into a coherent story. This results in causal fabrication: the invention of logical connectors to bridge failures in perception. In the top-left example in Figure 5, Gemini hallucinates a change in the static environment (ramp tilting) to explain the motion of the sphere going up the incline without any visible external force. Further analysis in (Chen et al., 2024a), shows that when models fail to maintain a true sequence across distant segments, they do not default to uncertainty but to hallucinated logical bridges. (Li et al., 2025d) confirms that this manifests as the invention of unobserved actions, such as claiming a person "opened" a door to explain a simple cut between rooms. A visualization of this failure mode is shown in Figure 6, where a ball miraculously appears on the other side of a solid wall after a camera pan. Rather than identifying this physical impossibility, the model fabricates a detailed mechanical explanation, claiming the impact "[causes] the entire wall to rotate clockwise... opening a path like a revolving door", an event that never occurs in the video. This narrative coherence bias is so overpowering that models actively restructure the timeline to ensure the video makes sense, even if it makes the answer wrong (Lee et al., 2025).

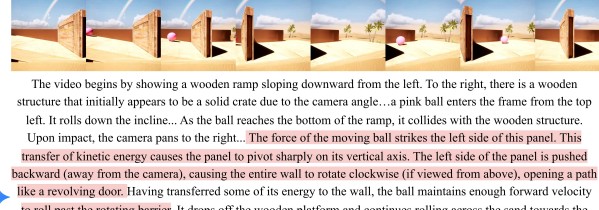

**Describe the complete sequence of motion and events in this video from start to finish. Focus specifically on the dynamics of the scene.**

The video begins by showing a wooden ramp sloping downward from the left. To the right, there is a wooden structure that initially appears to be a solid crate due to the camera angle…a pink ball enters the frame from the top left. It rolls down the incline... As the ball reaches the bottom of the ramp, it collides with the wooden structure. Upon impact, the camera pans to the right...The force of the moving ball strikes the left side of this panel. This transfer of kinetic energy causes the panel to pivot sharply on its vertical axis. The left side of the panel is pushed backward (away from the camera), causing the entire wall to rotate clockwise (if viewed from above), opening a path like a revolving door. Having transferred some of its energy to the wall, the ball maintains enough forward velocity to roll past the rotating barrier. It drops off the wooden platform and continues rolling across the sand towards the background...The panel continues to swing from the momentum of the impact. It rotates roughly 90 degrees before the friction and gravity of the simulation cause it to slow down. By the end of the video, the panel wobbles slightly as it comes to a rest in a new position, standing perpendicular to its original orientation, while the camera holds steady on the now-stationary structure.

*Figure 6.* **Causal Fabrication** In an IntPhys2 "teleportation" video (Bordes et al., 2025), Gemini-2.5-Pro explains an impossible transition (ball appearing to go through wall) by hallucinating an impact-driven wall rotation (red highlights).

## 4. Alternative Views

**Reliance on static semantics and world knowledge is not a failure but a rational and efficient design choice.** Many real-world video tasks are strongly predictable from objects, scene context, and affordances. We agree that actions correlate tightly with canonical objects, viewpoints, and affordances. From this perspective, treating video as a sparse set of informative frames is an effective approximation: it reduces computational cost, simplifies training, and exploits strong statistical regularities. If the goal is to answer "what is happening?" in the average case, static cues often suffice. Our position is not that models should ignore appearance, nor that priors are inherently harmful. Rather, we argue that predictiveness should not be conflated with understanding. Appearance-based shortcuts fail catastrophically in counterintuitive scenarios, from real-world safety anomalies (e.g., a car reversing unexpectedly) to AI-generated hallucinations that defy physics. In this sense, we do not reject the efficiency-oriented view; we delimit it. Static cues are a powerful first approximation. However, a system that cannot reliably abandon those shortcuts when they become misleading is not truly *video-aware*.

**Video-native encoders and scaling may solve the problem.** A reasonable counterposition is that the issue is temporary, stronger video-native encoders, larger context windows, denser sampling, or higher FPS should eventually recover the missing temporal signal. We agree that these directions are important and likely reduce the severity of many failures. Video-native encoders can encode temporal structure before the language bottleneck, and denser sampling can make brief transitions less likely to be skipped. However, these remedies make temporal evidence available, but do not by themselves make its use required. Existing diagnostic results summarized in Table 2 show that strong models can still fail when the decisive evidence is subtle, counterintuitive, or in conflict with learned event scripts. Thus, our claim is not an impossibility result against scaling, it is a requirement on the next paradigm. Improvements in representation should be paired with protocols that verify whether the answer depends on the relevant temporal evidence.

**If the LLM paradigm inherently biases models away from dynamics, we should drop the language backbone and revert to video-native architectures.** If the introduction of an LLM into video understanding models induces dynamic information loss, then the principled response is to remove the language backbone and revert to video-native architectures optimized for temporal credit assignment. We reject this as a false dichotomy. Our claim is not that LLMs are incompatible with video, but that current pipelines often allow language priors to dominate when spatiotemporal evidence is underspecified, yielding plausible answers without enforcing state tracking. At the same time, we see a highly promising trajectory for integrating LLM capabilities into computer vision: scalable supervision, compositional semantics, and interactive reasoning interfaces can become foundational to video understanding if they are coupled to verifiable perception. The path forward is therefore agentic and evidence-gated: treat the LLM as a controller that iteratively selects what to inspect (time windows, regions, tracked entities), requests motion-preserving signals (trajectories/flow/temporal tokens), and then validates or revises hypotheses via explicit consistency checks.

**Video-first tasks already evaluate dynamics.** Tracking, video object segmentation, temporal action localization, and motion segmentation directly measure aspects of pixel dynamics, so one might argue that the community already has the right evaluation tools. We agree that these tasks provide essential grounding principles. However, they are often isolated from Video LLM evaluation, where success is measured through language outputs that may not require the model to expose trajectories, masks, or temporal evidence. Our position is to import the discipline of video-first evaluation into language-mediated video understanding. When a model answers a video question, or summarizes an even, the claim should be verifiable against localized spatiotemporal evidence rather than accepted only as plausible text.

## 5. Call to Action: Towards Dynamically-Aware Video Understanding

Progress in video understanding will not come from further scaling context windows or language-model capacity alone, but from architectural, algorithmic, and evaluative shifts that treat spatiotemporal dynamics as first-class evidence rather than optional context.

**Motion-preserving Representations.** Current video-language models rely heavily on coarse tokenization schemes that discard high-frequency motion information before it reaches the semantic bottleneck. We argue that the fundamental unit of video understanding should not be static objects, but *transformations over time*. This requires representations that explicitly encode temporal change and directionality, including derivatives of the visual signal that preserve motion structure (Bagad & Zisserman, 2025). Recent work on pixel-dense embeddings that distill optical flow into high-resolution feature grids represents a step in this direction, enabling dynamic information to be retained prior to semantic abstraction (Araslanov et al., 2025). The input representation should preserve transformations over time rather than only static objects. Practical implementations include trajectory tokens, optical flow, or point-tracking derived features, temporal derivatives of object boxes, and video-native features that retain direction, speed, and state change before visual tokens are compressed into the LLM space. The key design principle is that motion should not be reconstructed only after semantic abstraction; it should remain accessible as a first-class signal during generation.

**Structurally Enforced Spatiotemporal Grounding.** Models cannot be expected to prioritize dynamics if their architectures allow visual evidence to be bypassed in favor of stronger language priors. We therefore advocate for structurally enforced grounding mechanisms in which generation is explicitly constrained by spatiotemporal visual evidence. Frameworks such as PerceptionLM (Cho et al., 2025) demonstrate the value of perception encoders that require fine-grained, temporally grounded descriptions. Complementary approaches, including self-diagnosis and contrastive verification (Huang et al., 2025), further reduce hallucination by penalizing reliance on priors and encouraging consistency with counterfactual visual evidence. More broadly, replacing free-form textual outputs with pixel-aligned predictions—such as temporal segments or spatiotemporal masks (Deng et al., 2025; Bai et al., 2024)—ensures that dynamics, rather than narrative plausibility, become the discriminative signal.

**Evaluation and Benchmark Standards.** Finally, progress must be enforced through evaluation. We argue that benchmarks should satisfy three minimal requirements: (1) *temporal sensitivity tests*, such that performance degrades under frame shuffling, reversal, or temporal corruption; (2) *evidence localization*, requiring models to identify when in the video the supporting evidence occurs, not only what the answer is; and (3) *pixel-level verifiability*, whereby temporal and causal claims can be explicitly checked against observable motion and state changes. Benchmarks that fail to enforce these constraints permit success via static cues and learned scripts, rewarding narrative fluency rather than genuine temporal percep-

tion. Recognizing what did not happen—especially in counterfactual or anomalous scenarios—is as critical as recognizing what did, and should be treated as a first-class evaluation signal. Metrics should therefore report not only answer accuracy, but also localization accuracy, mask or track overlap when available, and consistency under temporal perturbation. These protocols directly connect Video LLM evaluation to video-first tasks such as tracking and segmentation, while preserving the language-mediated interface that makes Video LLMs useful.

## 6. Conclusion

The pixels already contain the answer. When models succeed while overlooking what visibly moves and changes, video understanding becomes detached from video itself. Video LLMs must not ignore the pixel dynamics in plain sight: models and benchmarks must make spatiotemporal evidence unavoidable, and must fail when motion and state transitions are ignored or contradicted. Only then can progress reflect genuine perception rather than fluent but ungrounded narration.

## Impact Statement

This position paper argues for evaluation and modeling practices that make spatiotemporal grounding a requirement for Video LLMs. If adopted, the positive impact is improved reliability of video-based decision support by reducing ungrounded temporal claims. These failures are especially pronounced in safety-critical settings such as autonomous driving, healthcare monitoring, and security settings, where incorrect event ordering, missed state transitions, or fabricated explanations can lead to harmful and life-impacting events. By advocating for models and diagnostics that enforce temporal sensitivity and evidence localization, this work aims to steer the field toward models that fail safely, for example, by abstaining from uncertain predictions or explicitly requesting additional evidence.

A potential negative impact is that explicitly characterizing these failure modes may lower the barriers for adversaries to exploit deployed systems. To mitigate this risk, we encourage researchers and practitioners to incorporate the diagnostic recommendations proposed in this paper into both pre-release evaluation and continuous monitoring pipelines, helping ensure robustness, accountability, and trust as video-based AI systems continue to advance.

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
