# OpenReview forum: "Position: Video LLMs Must Not Ignore the Pixel Dynamics in Plain Sight"
_ICML.cc/2026/Position_Paper_Track — ICML 2026 Position Paper Track regular_

### Official Review · Reviewer_2R7w · 2026-02-17

**Significance:** 3
**Argument Clarity:** 3
**Rating:** 4
**Confidence:** 3

**Questions:**

- Do the identified failures differ in severity between CLIP-based encoders and video-native encoders such as VideoMAE?
- How much do the demonstrated failures diminish with increased frame sampling? For example, 32 or 64 frames?

**Alternative Views Section:**

Yes

**Compliance With Llm Reviewing Policy A Conservative:**

Affirmed.

**Discussion Potential:**

3

**Final Justification:**

Rebuttal resolved my concerns. Raising my score. Please add concrete protocols and broader model evaluation in the revision.

**Paper Summary:**

The paper argues that: current Video LLMs rely on static cues + language priors more than temporal perception. Then it synthesizes diagnostic evidence into a failure taxonomy. Finally, it calls for enforced spatiotemporal grounding in models and benchmarks.

**Position:**

Yes

**Position In Title:**

Yes

**Related Work:**

4

**Strengths And Weaknesses:**

Strengths:
- This paper raises a gap between benchmark performance and actual temporal perception in Video LLMs.
- Table 2 is very well elaborated and maps around 17 diagnostic probes into two failure modes.
- This paper analyzes the problem from two different angles: from both benchmark and architecture perspectives, and shows that the problem is rooted in evaluation practices and design choices simultaneously.

Weaknesses:
- The three directions in Section 5 lack concrete and actionable protocols. For example, what annotations, metrics, or infrastructure are needed?
- Current experiments are limited to 100 AVoE clips. Moreover, all the evaluations are based on a single model (Gemini-2.5-Pro). More diverse evidence is needed to support the main claim.
- The paper does not systematically map existing approaches, such as PerceptionLM, Video-R1, etc., onto the failure taxonomy in order to identify the remaining unsolved issues.

**Support:**

2

---

> ### Author Rebuttal · Authors · 2026-03-31
>
> We thank the reviewer for their review. We are encouraged that they recognized the paper’s central intent. We address their concerns below.
>
> **W1.** We agree that Section 5 would benefit from more explicit protocols, and we can clarify this in revision. We argue that visual grounding can be realized through spatiotemporal supervision, such as timestamped evidence spans and tracked bounding boxes. Recent works beginning to adopt the explicit verification we argue for include PerceptionLM, which emphasizes detailed perception supervision [1], Motion-Grounded Video Reasoning, which requires spatiotemporal segmentation-mask answers [2], and Open-o3 Video, which grounds reasoning with explicit timestamps and bounding boxes [3].
>
> **W2.** We agree that our own experiments are small in scope. Our goal was to use limited probes as illustrations while grounding the main argument in the broader taxonomy in Table 2. The point of that taxonomy is precisely that these papers collectively tell a consistent larger story when organized into the two persistent failure modes we identify. We also kept our illustrative evaluation aligned with the strongest frontier models available at the time: VidHalluc [4] (Table 3), Vinoground [5] (Table 2), and Favor-Bench [6] (Table 2) all report top-performing scores from Gemini-1.5 Pro and/or GPT-4o.
>
> **W3.** We agree this would strengthen the paper, and we can make it clearer in revision.  PerceptionLM addresses fine-grained perceptual grounding, directly pushing against static-cue dominance. Video-R1 [7] addresses the reasoning side through RL, but does not itself guarantee verifiable spatiotemporal grounding. Our holistic view is that many recent approaches address one side of the problem, but the unsolved issue is their integration under evaluations that verify reliance on true temporal and visual evidence.
>
> **Q1.** Yes, but current evidence suggests the issue is attenuated rather than eliminated. CLIP-based encoders are especially vulnerable because they leave temporal integration to later modules or the LLM; recent work explicitly identifies this as a source of motion errors and hallucination [8]. Video-native encoders such as VideoMAE help, but pretraining on coarse action labels still leaves a gap for fine-grained, continuous motion and subtle temporal change.
>
> **Q2.** Recent work shows that better frame selection or larger frame counts can improve benchmark scores [9]. This is also consistent with our qualitative probes: in Figures 4 and 5, Gemini-2.5 Pro still makes fundamental temporal errors even when sampled at 8 fps. Thus, sparse sampling can worsen the problem, but the persistence of errors even under denser context suggests that current models still struggle to use that evidence reliably once it is present.
>
> [1] Cho, J. H., Madotto, A., Mavroudi, E., Afouras, T., Nagarajan, T., Maaz, M., Song, Y., Ma, T., Hu, S., Jain, S., Martin, M., Wang, H., Rasheed, H., Sun, P., Huang, P.-Y., Bolya, D., Ravi, N., Jain, S., Stark, T., Moon, S., Damavandi, B., Lee, V., Westbury, A., Khan, S., Krähenbühl, P., Dollár, P., Torresani, L., Grauman, K., and Feichtenhofer, C. PerceptionLM: Open-access data and models for detailed visual understanding, 2025.
>
> [2] Deng, A., Chen, T., Yu, S., Yang, T., Spencer, L., Tian, Y., Mian, A. S., Bansal, M., and Chen, C. Motion-grounded video reasoning: Understanding and perceiving motion at pixel level. In Proceedings of the IEEE/CVF Conference on Computer Vision and Pattern Recognition (CVPR), pp. 8625–8636, June 2025.
>
> [3] Meng, J., Li, X., Wang, H., Tan, Y., Zhang, T., Kong, L., Tong, Y., Wang, A., Teng, Z., Wang, Y., et al.: Open-o3 video: Grounded video reasoning with explicit spatio-temporal evidence. arXiv preprint arXiv:2510.20579, 2025.
>
> [4] Li, C., Im, E. W., and Fazli, P. Vidhalluc: Evaluating temporal hallucinations in multimodal large language models for video understanding. In Proceedings of the IEEE/CVF Conference on Computer Vision and Pattern Recognition (CVPR), pp. 13723–13733, June 2025a.
>
> [5] Zhang, J., Cai, M., and Lee, Y. J. Vinoground: Scrutinizing LMMS over dense temporal reasoning wth short videos, 2024. URL https://arxiv.org/abs/2410.02763.
>
> [6] Tu, C., Zhang, L., Chen, P., Ye, P., Zeng, X., Cheng, W., Yu, G., and Chen, T. Favor-bench: A comprehensive benchmark for fine-grained video motion understanding. arXiv preprint arXiv:2503.14935, 2025.
>
> [7] Feng, K., Gong, K., Li, B., Guo, Z., Wang, Y., Peng, T., Wu, J., Zhang, X., Wang, B., and Yue, X. Video-r1: Reinforcing video reasoning in mllms. arXiv preprint arXiv:2503.21776, 2025b.
>
> [8] Ding, Xi, and Lei Wang. "Do language models understand time?." Companion Proceedings of the ACM on Web Conference 2025. 2025.
>
> [9] Li, Y., Tang, C., Zhuang, J., Yang, Y., Sun, G., Li, W., Ma, Z., and Zhang, C. Improving LLM Video Understanding with 16 Frames Per Second. In Proceedings of the Forty-second International Conference on Machine Learning (ICML), 2025.

---

> > ### Author Rebuttal · Reviewer_2R7w · 2026-04-03
> >
> > Thank the authors for the rebuttal. Concrete examples address W1, a taxonomy-centered argument justifies W2, and method-to-failure mapping resolves W3. My concerns are all addressed, and I am willing to raise my score.

---

### Official Review · Reviewer_QAMY · 2026-03-03

**Significance:** 4
**Argument Clarity:** 4
**Rating:** 5
**Confidence:** 5

**Questions:**

Please refer to the weaknesses.

**Alternative Views Section:**

Yes

**Compliance With Llm Reviewing Policy A Conservative:**

Affirmed.

**Discussion Potential:**

4

**Ethical Review Concerns:**

N/A.

**Final Justification:**

I think the authors' rebuttal has addressed my concerns, so I keep my original positive opinion about this work.

**Paper Summary:**

This paper proposes a position that recent progress achieved by Video Large Language Models (VideoLLMs) is measured by benchmarks and protocals that cannot truly reflect the models' understanding capabilities of spatiotemporal evidence but largely dominated by shortcuts, i.e., the static cues and language priors. Specifically, the paper first analyzes reasons why current models suceed at benchmarks while failing at elementary temporal perception from two perspectives of benchmark and architecture. Then it demonstrates a lot of examples to illustrate the two failure modes of current VideoLLMs and also summarizes the reported pathologies from recent diagnostic methods into a detailed table. In addition, the paper lists several credible alternative views and gives some critical discussions regarding them in the end of the paper.

**Position:**

Yes

**Position In Title:**

Yes

**Related Work:**

4

**Strengths And Weaknesses:**

Strengths:

1.The paper is well-written and easy to understand.

2.The position that current benchmarks and evaluation protocals cannot truly measure the progress of VideoLLMs in perceiving temporal information is important.

3.The paper provides a large number of visualization results and cited diagnostic methods to illustrate the failure modes of current VideoLLMs, which makes the content highly readable and also helpful to peer researchers in related areas for review.

Weaknesses:

1.In the current manuscript, it seems that more up-to-date frontier models like Qwen3-VL, GPT-5o and Gemini-3 Pro are missing. How would these more advanced models behave for temporal perception and what would be their failure modes are still unclear.

2.A suggestion is to strengthen the contribution of this paper by adding some potential technical solutions for addressing the vulnerability of current VideoLLMs in perceiving fine-grained temporal information or correcting the existing issues in current evaluation suite.

**Support:**

3

---

> ### Author Rebuttal · Authors · 2026-03-31
>
> We thank the reviewer for the very encouraging review and are grateful they recognized several aspects of the paper that were central to our intent. We appreciate the recognition that the paper contributes a detailed synthesis of recent work on the failure taxonomy, with the visual examples and Table 2 intentionally designed to make existing methods more useful for mitigating these failures in future research directions.
>
> **W1.** For further elaboration on this point, we also refer to the related concern raised in **W2** by Reviewer **2R7w**. Here, our evaluation was kept consistent with the frontier proprietary baselines used throughout the benchmark papers in Table 2, while updating to Gemini-2.5 Pro as the strongest model available to us at the time, rather than the commonly reported Gemini-1.5 Pro. The aim was to test whether the identified failures remain visible even at the closed-source frontier, and the qualitative examples in Figures 2-5 suggest that they do. We view the models you mentioned (e.g., Qwen3-VL, GPT-5o, Gemini-3 Pro) as an exciting direction for further validation. Based on current trends, we expect improvements in overall reasoning and multimodal alignment, but core challenges around temporal grounding and motion understanding still persist. In a revision, we would be happy to clarify this positioning and discuss these newer models more explicitly as part of the forward-looking perspective.
>
> **W2.** We agree, and this was precisely the motivation for Section 5. Our intention was to move beyond diagnosis and outline technical directions, though we also agree that these can be made more explicit in the write-up to strengthen the contribution and make clear directions forward for researchers. In particular, Section 5 was meant to point toward motion-preserving representations, structurally enforced spatiotemporal grounding, and evaluation standards that require temporal sensitivity, evidence localization, and pixel-level verifiability. We are open to clarifying these directions more concretely in revision.

---

> > ### Author Rebuttal · Reviewer_QAMY · 2026-04-04
> >
> > I have no more concerns after reading the authors' rebuttal. And I strongly encourage the additional results and discussions can be added to the revision of current manuscript.

---

### Official Review · Reviewer_9kSB · 2026-03-13

**Significance:** 3
**Argument Clarity:** 3
**Rating:** 5
**Confidence:** 3

**Questions:**

1. Can the authors provide more systematic empirical evidence beyond the qualitative examples?

**Alternative Views Section:**

Yes

**Compliance With Llm Reviewing Policy A Conservative:**

Affirmed.

**Discussion Potential:**

3

**Final Justification:**

My concerns are addressed, thus I keep the posive score.

**Paper Summary:**

This paper argues that current Video LLMs give the illusion of temporal understanding because benchmarks and architectures allow them to perform well without truly using spatiotemporal evidence. Accordingly, the authors identify two coupled failure modes: static-cue dominance and prior‑driven temporal hallucination, and argue that progress must re-center on pixel dynamics via motion-preserving representations.

**Position:**

Yes

**Position In Title:**

Yes

**Related Work:**

3

**Strengths And Weaknesses:**

Strengths:
1. The authors provide an unusually broad and up-to-date survey of relevant work.
2. The proposed pixel dynamics in plain sight is conceptually novel and aligns with many practitioners' informal impressions.


Weaknesses:
1. The position rests heavily on illustrative examples rather than controlled experiments or empirical results that support the claims.
2. Pixel dynamics is not well defined and precisely delimits tasks where this is a hard requirement versus tasks where using priors and sparse frames is entirely acceptable.
3. The paper lacks the discussion of the trade-offs and efficiency constraints.

**Support:**

3

---

> ### Author Rebuttal · Authors · 2026-03-31
>
> We thank the reviewer for the thoughtful and supportive assessment. We are happy to see that the reviewer recognized both the breadth of the survey and the central conceptual contribution of the paper. Specifically, that important temporal evidence can remain visible in the pixels yet still be underused by current Video LLMs.
>
> **W1.**  We agree that our own experiments are intentionally limited and are not meant to serve as the sole empirical basis for the claim. The main evidentiary basis of the paper is the broader body of recent diagnostic work synthesized in Table 2; our qualitative and small quantitative examples are included as concrete illustrations of the same recurring pattern. We can make this clearer in the write-up.For a concrete example, For a concrete quantitative example, TVBench [1] measures the spatial bias of MVBench by comparing model performance when given only a single image, a shuffled video, or the full video. As shown in Table 3 of that paper, performance remains similar even when temporal order is destroyed. GPT-4o scores 62.0 with only an image, 68.5 on shuffled videos, and only 66.7 on the full video. Likewise, Gemini 1.5 rises only from 57.2 (image) to 67.3 (video). This is exactly the kind of controlled evidence underlying our position. By bringing many such probes together in one taxonomy, we show that they converge and support our claims.
>
> **W2.**  By pixel dynamics, we mean the information carried in the transition between frames: how a specific object or subject evolves over time within a scene, its motion, direction, state change, or causal interaction. It is the part of the visual signal that cannot be resolved from a single frame alone, cannot be reduced to static appearance, and should not be recoverable purely from language priors. We claim that when a benchmark is presented as measuring temporal understanding, the correct answer should depend on this between-frame evidence and the model’s reliance on it should be verifiable. We can revise the paper to make this definition and boundary much clearer
>
> **W3.** We agree this deserves stronger discussion. Our intent in the Alternative Views section was precisely to engage the assumptions motivating current architectures, namely that static semantics often suffice, sparse temporal processing is computationally attractive, and language backbones can compensate for incomplete perception. Our position is not anti-efficiency; rather, it is that efficiency should not be conflated with genuine temporal understanding when the task depends on verifiable dynamics. We can make this trade-off discussion more explicit in revision.
>
> **Q1.**  Thank you for this helpful suggestion; we agree that adding more explicit empirical grounding would further strengthen the paper. Our intent was to synthesize and reinterpret the growing body of systematic evidence already reported across recent works, rather than introduce a new benchmark within the scope of a position paper. Many of these studies consistently point to the same patterns we highlight, particularly the reliance on static cues and limited temporal grounding. In our revision, we will make this connection more explicit and clarify how the qualitative examples reflect broader, well-established empirical trends across the literature.

---

> > ### Author Rebuttal · Reviewer_9kSB · 2026-04-03
> >
> > My concerns are addressed.

---

### Official Review · Reviewer_CH7F · 2026-03-14

**Significance:** 3
**Argument Clarity:** 2
**Rating:** 4
**Confidence:** 5

**Questions:**

Q1.  Is static-cue dominance a failure mode, or a necessary prerequisite for temporal understanding? If video models are assumed to learn in a coarse-to-fine manner, scene-level understanding first and then temporal dynamics, the current static-cue dominance could be an intermediate step toward perfect understanding of temporal dynamics rather than a fundamental flaw.

Q2. Does context token limit or frame subsampling contribute to the loss of temporal dynamics in Video LLMs?

**Alternative Views Section:**

Yes

**Compliance With Llm Reviewing Policy A Conservative:**

Affirmed.

**Discussion Potential:**

2

**Final Justification:**

After reviewing the rebuttal and follow-up discussions, I am raising my score to Borderline Accept. My core concern is that the proposed failure modes are well-known in video AI research (W1). After discussion, the authors clarified their core position: they do not claim the failure modes are new, but but rather that the VLM structurally normalizes them. Additionally, the authors provided a concrete plan to restructure the "Alternative Views" section (W3) to  discuss scaling, higher FPS, and video-native encoders.

I note that my final recommendation is conditional, expecting that the authors faithfully revise the manuscripts to incorporate the clarifications discussed during the rebuttal.

**Paper Summary:**

This position paper argues that current video LLMs are fundamentally constrained by two failure modes: static-cue dominance, where models rely on appearance and contextual signals rather than spatiotemporal evidence, and prior-driven temporal hallucination, where learned regularities override observed pixel dynamics. For example, widely-used benchmarks can be solved without processing temporal information, and architectural design systematically defer temporal understanding to the LLM backbone. In addition, the video LLMs often ignore the temporal dynamics in a video if the movement is unintuitive or subtle. The paper calls for the community to consider spatiotemporal evidence an prior requirement in both model design and benchmark construction.

**Position:**

Yes

**Position In Title:**

Yes

**Related Work:**

3

**Strengths And Weaknesses:**

**Strengths**

S1. The paper position targets an important and practical problem in video understanding research.

S2. The paper provides a comprehensive survey of existing benchmarks for video LLMs to diagnose the capability to understand pixel dynamics over static-cue dominance and prior-driven temporal hallucination.

**Weaknesses**

W1. Although static-cue dominance and prior-driven temporal hallucination are well-known and long-standing problems in video understanding research, the paper frames these failures specific to video LLMs. In addition, this paper confirms the same issues persist in video LLMs as many references in this paper have claimed instead of providing a new position.

W2. Although the paper mentions  architectural causes (e.g., video encoders, fusion modules), the analysis remains superficial. The paper primarily cites existing work without providing a deeper structural analysis of why Video LLMs fundamentally fail at temporal dynamics.

W3. The alternative views in the paper are not sufficiently counterpositions. Although the efficiency is important part of video understanding models, I think no one in the community will agree that pixel dynamics can be ignored for efficiency. Also, alternative views, such as whether scaling video-native encoders could resolve the issue, or whether leveraging the reasoning capability of LLMs could compensate for weak temporal signals, are not discussed enough.

W4. The video understanding tasks in this paper are limited to QA-based benchmarks, but tasks that more directly measure pixel dynamics, such as object tracking or video object segmentation, are not discussed as potential evaluation directions.

W5.  The writing quality should be improved. Although this paper targets to claim a position, this paper follows a repetitive pattern of assertion → citation → next assertion without providing in-depth analysis and explanations.

**Support:**

3

---

> ### Author Rebuttal · Authors · 2026-03-31
>
> We thank the reviewer for taking the time to read and evaluate our position paper. We appreciate the recognition that the paper targets an important problem and provides a comprehensive synthesis of existing benchmarks and diagnostics. We address the reviewer’s questions and concerns below.
>
> **W1.** We agree that static cue reliance is not a new problem invented by Video LLMs, nor do we claim that temporal bias was absent from earlier video understanding research. Our position is that Video LLMs change the failure mode structurally by coupling weak or compressed visual-temporal evidence with a powerful pretrained language model that can complete a plausible event narrative even when the video evidence is contradictory. In earlier video understanding pipelines, predictions were more tightly tied to video-native supervision (e.g., action classification, object tracking). In Video LLMs, static cues and priors become a persistent substitute for genuine temporal grounding. Our claim is not that these pathologies are new, but that Video LLMs create a regime in which they are systematically normalized by current protocols. This is also why we frame the issue as a warning about a precedent we risk standardizing. This concern is increasingly important as video interaction moves toward language-based settings (e.g., search, summarization) where users will naturally trust textual outputs.
>
> **W2.** We agree that our paper does not provide a new interpretability study of all architectural components of Video LLMs. Our aim is not to isolate a novel component-level flaw in one model, but to synthesize a growing body of evidence into a position on the direction current architectures are taking us: temporal evidence is increasingly weakened, compressed, or deferred before generation, while the language model remains capable of producing a coherent event narrative regardless. Section 2.2 in the paper is meant as a structural synthesis of existing findings. Our goal is to argue that these isolated analyses collectively point to a broader paradigm-level issue.
>
> **W3.** Our intention in the Alternative Views section was to engage with the arguments motivating the prevailing direction of Video LLM research. We agree, however, that the section could more directly address the reviewer’s two alternatives: whether sufficiently scaled video-native encoders could resolve much of the issue, and whether LLM reasoning can compensate for weak temporal signals. Our view is that these are promising directions, but the evidence surveyed in Table 2 suggests they are not sufficient alone. Failures persist even in stronger proprietary models. We can revise the section to make it clearer that stronger encoders and reasoning may help, but without mechanisms and evaluations that verify reliance on temporal evidence, they may still produce more fluent yet insufficiently grounded video understanding.
>
> **W4.** We agree that object tracking, video object segmentation, and other video-first tasks are highly relevant. Our focus on QA-based benchmarks was deliberate because the paper is specifically about Video LLMs as language-mediated interfaces to video (e.g., querying, summarization). In this method, the failure cases are especially important because they can be obscured by fluent textual output (see a visual explanation of this in Fig. 1 in the paper). At the same time, we agree that these video-first tasks should be discussed more explicitly as complementary directions. In fact, they are valuable precisely because they enforce tighter coupling to the video signal itself and therefore offer useful grounding principles for evaluating and improving Video LLMs. In that sense, they are not outside our position; rather, they provide examples of the kind of video-native grounding policies that are consistent with our call in Section 5 for pixel-aligned outputs such as temporal segments and spatiotemporal masks, rather than relying only on free-form text generation.
>
> **W5.** Thank you for this feedback. Our goal is to synthesize many recent results into a coherent position. In revision, we will make the causal links between ideas more explicit, reduce repetition, and improve the flow so the paper reads more clearly as a single argument.
>
> **Q1.** Static cues are necessary, but they become a failure mode when they substitute for, rather than support, temporal reasoning. Our concern is not coarse-to-fine processing itself; it is when the model never meaningfully progresses beyond the coarse static stage and remains insensitive to motion, order, and state change when those are decisive.
>
> **Q2.** Yes. Context limits and frame subsampling both contribute by sparsifying and compressing the video signal before reasoning, making temporal dynamics easier to lose. But the broader problem is that current Video LLMs can still produce plausible answers after this loss by falling back on static cues and language priors instead of grounding in temporal evidence.

---

> > ### Author Rebuttal · Reviewer_CH7F · 2026-04-04
> >
> > I appreciate the authors' rebuttal. However, I find that it does not provide sufficiently concrete rationales to fully address my concerns. I therefore have the following questions for further clarification:
> >
> > W1. The rebuttal claims that video LLMs change the failure mode "structurally" due to their language-based interface. However, in prior action recognition research, although action labels are framed as classification targets, each label inherently encodes rich semantic information of textual form of actions. From this perspective, observing similar failure modes (e.g., scene bias) in video LLMs does not appear fundamentally new. Could the authors provide a more "systematic" analysis comparing the failure modes in traditional video understanding methods and those in Video LLMs? As the current manuscript focuses almost exclusively on video LLMs, it is unclear whether researchers familiar with the history of video understanding would consider this a novel contribution.
> >
> > W3. Could the authors clarify how Table 2 supports the claim that they are insufficient? In particular, the manuscript does not discuss directions such as video-native encoders or increasing FPS, but both of which have demonstrated substantial improvements in motion understanding for Video LLMs (e.g., Gemini-3 with fps=10). Additionally, could the authors elaborate on how they plan to revise the Alternatives section? As a position paper, the section, Alternative Views, is critical, yet the current discussion appears largely speculative and lacks concrete justification (e.g., statements in rebuttal such as “may still produce ...”).
> >
> > Q1. Since a model learns a scene (frame) information first and then temporal dynamics, the current failure modes can be due to insufficient performance of video LLMs. For example, increasing computes by higher fps in video LLMs' subsampling significantly improves the accuracy of temporal understanding. If the proposed failure modes are fundamental issues, I think the authors should sufficient rationales to claim that the video LLMs should fail even after grounding the temporal information in a video due to language-based interface and learned knowledge.
> >
> > Q2. If increasing context length or sampling higher FPS (e.g., 10 FPS in systems such as Gemini) continues to improve performance, do these failure modes still persist due to a lack of true pixel-level temporal understanding? Furthermore, could reasoning mechanisms such as chain-of-thought (CoT) provide sufficient temporal grounding within the current paradigm?

---

### Decision · Program_Chairs · 2026-04-30

**Decision:**

Accept (regular)

**Comment:**

The reviews consistently acknowledge the paper’s contribution and its positioning. The work addresses an important and practical problem in video understanding research. The authors have also successfully addressed several of the remaining concerns, strengthening the case for acceptance.